# Elevated Copeptin Levels Are Associated with Heart Failure Severity and Adverse Outcomes in Children with Cardiomyopathy

**DOI:** 10.3390/children10071138

**Published:** 2023-06-30

**Authors:** Karan B. Karki, Jeffrey A. Towbin, Samir H. Shah, Ranjit R. Philip, Alina N. West, Sachin D. Tadphale, Arun Saini

**Affiliations:** 1Division of Pediatric Cardiology, Department of Pediatrics, Le Bonheur Children’s Hospital, University of Tennessee Health Science Center, Memphis, TN 38103, USA; jtowbin1@uthsc.edu (J.A.T.); rphilip@uthsc.edu (R.R.P.); sachintadphale@gmail.com (S.D.T.); 2Division of Pediatric Critical Care Medicine, Department of Pediatrics, Le Bonheur Children’s Hospital, University of Tennessee Health Science Center, Memphis, TN 38103, USA; sshah7@uthsc.edu (S.H.S.); awest3@uthsc.edu (A.N.W.); 3Section of Pediatric, Critical Care Medicine, Department of Pediatrics, Baylor College of Medicine and Affiliated Texas Children’s Hospital, Baylor College of Medicine, Houston, TX 77030, USA; asaini@bcm.edu

**Keywords:** copeptin, cardiomyopathy, pediatric heart failure, BNP, biomarkers, neurohormonal

## Abstract

In children with cardiomyopathy, the severity of heart failure (HF) varies. However, copeptin, which is a biomarker of neurohormonal adaptation in heart failure, has not been studied in these patients. In this study, we evaluated the correlation of copeptin level with functional HF grading, B-type natriuretic peptide (BNP), and echocardiography variables in children with cardiomyopathy. Furthermore, we determined if copeptin levels are associated with adverse outcomes, including cardiac arrest, mechanical circulatory support, heart transplant, or death. In forty-two children with cardiomyopathy with a median (IQR) age of 13.1 years (2.5–17.2) and a median follow-up of 2.5 years (2.2–2.7), seven (16.7%) children had at least one adverse outcome. Copeptin levels were highest in the patients with adverse outcomes, followed by the patients without adverse outcomes, and then the healthy children. The copeptin levels in patients showed a strong correlation with their functional HF grading, BNP level, and left ventricular ejection fraction (LVEF). Patients with copeptin levels higher than the median value of 25 pg/mL had a higher likelihood of experiencing adverse outcomes, as revealed by Kaplan–Meier survival analysis (*p* = 0.024). Copeptin level was an excellent predictor of outcomes, with an area under the curve of 0.861 (95% CI, 0.634–1.089), a sensitivity of 86%, and a specificity of 60% for copeptin level of 25 pg/mL. This predictive value was superior in patients with dilated and restrictive cardiomyopathies (0.97 (CI 0.927–1.036), *p* < 0.0001, *n* = 21) than in those with hypertrophic and LV non-compaction cardiomyopathies (0.60 (CI 0.04–1.16), *p* = 0.7, *n* = 21).

## 1. Introduction

Heart failure in children is associated with significant morbidity and mortality [1,2]. Cardiomyopathies are rare but important causes of heart failure in children [3,4,5]. Pediatric cardiomyopathies are a heterogeneous group of disorders. The different types of cardiomyopathies are dilated, hypertrophic, left ventricular non-compaction, and restrictive cardiomyopathies [3]. The overall incidence of pediatric cardiomyopathies is estimated at about 1.1 to 1.5 per 100,000 children [3]. Among the different forms of cardiomyopathy, dilated cardiomyopathy is the most common type, followed by hypertrophic cardiomyopathy. Cardiomyopathies are the most common indication for heart transplantation in children over 1 year of age. Children with cardiomyopathy can have a variable severity of heart failure [5].

In addition to cardiomyopathies, heart failure in children can be caused by various diseases, including congenital structural heart defects, arrhythmias, toxins, and non-cardiac etiologies like sepsis [2,5]. While many of the causes mentioned above can be treated surgically or medically to reverse the underlying pathophysiological process, in contrast, cardiomyopathies, particularly dilated and restrictive cardiomyopathies, gradually lead to depressed systolic and diastolic function, resulting in end-stage heart failure despite medical management [2,4,5,6]. Patients with cardiomyopathies are at a high risk of adverse events, including cardiac arrest and death; therefore, close monitoring of their disease progression is vital. Monitoring using clinical assessment and biomarkers can help identify patients at risk of clinical deterioration to provide timely evaluations for heart transplantation and mechanical circulatory support like left ventricular assist devices. Novel neurohormonal biomarkers may aid in monitoring the disease progression in this high-risk cohort [7].

Neurohormonal activation plays a pivotal role in the progression of heart failure, and it also correlates with the severity of heart failure [6,7,8]. The sympathetic system and renin–angiotensin–aldosterone system (RAAS) are well-known pathways of neurohormonal changes which are associated with the severity of heart failure [8,9,10,11]. It is well established that elevated norepinephrine and angiotensin II are associated with poor prognosis in heart failure, and neurohormonal antagonism has improved survival in adults with heart failure [8,9,10,11]. In addition to these two well-established pathways, the arginine vasopressin (AVP) system is also thought to be a major neurohormonal process that plays an integral role in the progression of heart failure [12,13]. There has been significant clinical interest in AVP-related ventricular dysfunction and hyponatremia and the possible benefit of AVP antagonism [14,15].

AVP is a nonapeptide secreted by supraoptic and paraventricular nuclei in the hypothalamus [15,16]. It is released by the pituitary gland in the bloodstream in response to osmotic and non-osmotic stimuli. It acts in the vasopressin receptors (V1 and V2). The V2 receptors are in the collecting tubules of the kidneys. The activation of V2 receptors results in the retention of free water. Similarly, the V1 receptors are in the vasculature. The activation of V1 receptors leads to vasoconstriction resulting in an increase in systemic vascular resistance [15,16]. In a normal physiologic state, AVP elevation does not have significant hemodynamic effects, whereas, in patients with heart failure, AVP elevation can have significant effects on hemodynamics and ventricular function [16]. It has been postulated that AVP release stimulus is more shifted to a non-osmotic trigger compared to an osmotic trigger in heart failure. In patients with heart failure, V2 receptor activation leads to an increase in free water retention, which leads to an increase in preload. Similarly, V1 receptor activation leads to vasoconstriction leading to an increase in afterload to the left ventricle. In animal studies, AVP has been shown to act on the myocardium leading to myocardial hypertrophy [17]. Elevated AVP has been shown to be a marker of poor prognosis in the heart failure population and can be a potential prognostic biomarker [7,12,13].

Currently, AVP measurement has been limited in the experimental laboratory. AVP is an unstable molecule, predominantly bound to platelets and rapidly cleared due to its short half-life. Due to these characteristics, it cannot be measured by routine sandwich assays [18,19]. Copeptin, a C terminal part of the AVP precursor peptide, preprovasopressin, has been considered a surrogate of AVP. Copeptin is a larger molecular fragment of pre-pro vasopressin (a precursor molecule for AVP and copeptin) secreted in equimolar amounts to Arginine vasopressin (AVP) [18,19]. In contrast to AVP, copeptin is highly stable and reliably measured in stored plasma [18,19]. Therefore, copeptin has been used as a surrogate marker of AVP to assess neurohormonal adaptation in adults with heart failure [20,21,22,23].

Several adult studies in patients with heart failure have shown that copeptin can be a useful prognostic biomarker for acute and chronic heart failure [20,21,22,23]. Elevated copeptin levels have been associated with an increase in disease severity and mortality in critically ill children [24,25,26]. In a single-centered study, copeptin levels were associated with adverse outcomes in children with heart failure encompassing mostly children with congenital heart defects [26]. We conducted this prospective cohort study to measure copeptin levels in children with cardiomyopathy in different stages of heart failure. We assessed the correlation of copeptin level with established markers of the severity of heart failure, including functional heart failure grading, BNP level, and echocardiography (ECHO) variables of left ventricular function. Furthermore, we determined if copeptin levels are associated with adverse outcomes during the follow-up period.

## 2. Methods

In this single-center prospective cohort study, we enrolled children with cardiomyopathy in different stages of heart failure aged 1 month to 18 years at the Outpatient Heart Failure Clinic, Inpatient Wards, and Intensive Care Units of Le Bonheur Children’s Hospital, Memphis, Tennessee. We included children with underlying cardiomyopathy in whom the treating cardiologists diagnosed heart failure. We excluded children if they met any of the following exclusion criteria: age less than 1 month and more than 18 years, presence of acute renal injury (serum creatinine > 1.5 mg/dL), or presence of acute liver injury (Aspartate aminotransferase (AST)/Alanine aminotransferase (ALT) > 500 IU/dL). Healthy children were enrolled in the General Pediatric Outpatient Clinic of Le Bonheur Children’s Hospital. Informed consent was obtained from parents or legal guardians as per the University of Health Science Center (UTHSC) Institutional Review Board guidelines before the study enrollment.

We collected relevant demographic information, including age, gender, etiology of heart failure, medications (diuretic agents, vasoactive-inotropic medications, beta-adrenergic receptor blockers, angiotensin-converting enzyme inhibitors), respiratory support (high flow nasal cannula, non-invasive or invasive mechanical ventilation), and mechanical circulatory support (ECMO and VAD). The cases were assigned the functional heart failure grading based on either Ross heart failure grading (for children aged ≤5 years) or New York Heart Association (NYHA) heart failure grading (for children aged >5 years) by the treating cardiologists specialized in pediatric heart failure at the time of enrollment [27]. The treating cardiologists were blinded to the laboratory values. We drew 2 mL of blood for each subject in ethylenediaminetetraacetic acid (EDTA) tubes with aprotinin to measure copeptin level, BNP level, and basic metabolic panel within 24 h after enrollment. The supernatant plasma was centrifuged at 1600 revolutions per minute for 15 min at 4 °C and stored at −70 °C. Copeptin and BNP levels were analyzed in the UTHSC research laboratory using an ELISA method by the manufacturer’s protocol (Phoenix Pharmaceutical Inc. Burlingame, CA, USA). It costs USD 30 to test one sample of copeptin in our UTHSC research laboratory. The basic metabolic profile was used to calculate serum osmolality. We also obtained echocardiography data of the images performed within two days of the enrollment based on the clinical indication. The ECHO variables included left ventricular ejection fraction (LVEF), left atrial (LA) volume, and left ventricular end-diastolic dimension (LVEDD). A board-certified and blinded pediatric cardiologist independently read the echocardiograms. Each case was followed for a minimum of 2 years to record the occurrence of any adverse event. The composite adverse outcome was defined as cardiac arrest requiring cardiorespiratory resuscitation, worsening heart failure needing mechanical circulatory (ECMO or VAD), orthotopic heart transplant, or death.

## 3. Statistical Analysis

The study aimed to identify a 25% difference in copeptin levels between the cardiomyopathy and control groups. To achieve this, a sample size of 30 participants was chosen for each group, with a statistical power of 80% and an alpha of 0.05. Continuous variables were reported as medians and interquartile ranges (IQR), while categorical variables were summarized as frequency counts and percentages. A Wilcoxon signed-rank test was applied to compare the cases and the controls. Mann–Whitney U tests or Fisher’s exact tests were applied to compare the cases with and without adverse events. Spearman’s correlation coefficients were obtained to determine the correlation between copeptin level and established measures of the severity of heart failure. Receiver operator characteristics curves analyses were performed to determine the predictability of the composite adverse outcome by copeptin level, BNP level, and LVEDD. Kaplan–Meier survival curves were derived for the composite adverse outcomes among the cohort based on the median copeptin level in the cases. A significance level of *p* = 0.05 was used. Analyses were conducted using SPSS 24 (IBM Inc., Armonk, NY, USA).

## 4. Results

We enrolled 42 children with cardiomyopathy at different heart failure stages. Table 1 presents the clinical characteristics of the study group in detail. The cases in the study group for heart failure management were on various medications, including 50% (21) of cases on beta-blockers, 40% (17) on ACE inhibitors, 29% (12) on diuretics, and 26% (11) on vasoactive-inotropic meds. Many cases were on multiple medications, including 12% (5) on three or more agents, 33% (14) on two agents, 33% (14) on one agent, and 14 % (6) on no heart failure therapy at the time of enrollment. Additionally, 9% (4) cases were on mechanical ventilation. During the median (interquartile range) follow-up period of 2.3 years (2.0–2.6), 9% (4) of patients had cardiac arrest requiring cardiorespiratory resuscitation, 7% (3) needed mechanical circulatory support in the form of ventricular assist device, 14% (6) underwent an orthotropic heart transplant, and 2.4% (1) died. The composite adverse outcome event occurred in 16.6% (7) cases with a median time to adverse outcome of 101 days (2–675).

Children with cardiomyopathy who experienced adverse outcomes had higher copeptin levels (median (Interquartile Range (IQR)) of 741.8 pg/mL (351–9.14.1) compared to those without adverse outcomes (21 pg/mL (11.9–74.4), *p* = 0.045) and healthy children (6.2 pg/mL (5.8–6.7), *p* < 0.001), as shown in Figure 1. Copeptin levels were strongly positively correlated with functional heart failure grading (Rho 0.715, *p* < 0.001) and BNP level (Rho = 0.887, *p* < 0.001), as presented in Figure 2. Additionally, copeptin levels showed weak to moderate correlation with LA volume (Rho 0.600, *p* < 0.001), LVEF (Rho −0.442, *p* = 0.004), and serum sodium level (Rho −0.332, *p* = −0.005). However, no correlation was found between copeptin levels and patients’ age (Rho 0.037, *p* = 0.758) and serum osmolarity (Rho 0.103, *p* = 0.394), as presented in Table 2. Similarly, children with adverse outcomes had elevated BNP levels, higher LA volume, and lower LVEF compared to children without adverse outcomes, as presented in Table 1.

Univariate logistics regression analysis revealed that several factors, including copeptin level, functional HF grading, BNP level, LVEF, and LA volume, were associated with adverse outcomes. These findings are presented in Table 3. In addition, we determined that copeptin levels were independently associated with adverse outcomes even after adjusting for the severity of HF, as indicated by functional HF grading on the multivariable logistic regression analysis. Specifically, we found an adjusted Odds Ratio (OR) of 1.010 with a 95% CI of 1.002–1.019 and a *p*-value of 0.019.

The receiver operator characteristics (ROC) curves were obtained to assess predictive values of copeptin level, BNP level, LVEF, and LA volume for all cases, cases with dilated and restrictive cardiomyopathies, and cases with hypertrophic and LV non-compaction cardiomyopathies, as illustrated in Figure 3. Copeptin levels had moderate to high predictive value with the area under the curve (AUC) of 0.86 with a 95% CI of 0.63–1.08 and a *p*-value of 0.002 for all cases with a sensitivity of 86% and specificity of 60% for copeptin level of 25 pg/mL or higher. Copeptin levels had an even better predictive value (AUC 0.97 (95% CI 0.9–1.03), *p*-value < 0.001) in cases with either dilated and restrictive cardiomyopathy, but no predictive value in cases with either hypertrophic or LV non-compaction cardiomyopathy (AUC 0.61 (95% CI 0.05–1.17), *p*-value = 0.71). The predictive value of BNP level was similar to copeptin level in all cases, cases with dilated and restrictive cardiomyopathies, and cases with hypertrophic and LV non-compaction cardiomyopathies. Only LVEF had a predictive value in those with either hypertrophic or LV non-compaction cardiomyopathy. Lastly, we performed Kaplan–Meier survival time to event analysis using the median copeptin level of 25 pg/mL for all cases as the grouping variable. The log-Rank test showed a higher likelihood of adverse events in cases with copeptin levels higher than the median value during the follow-up period, as illustrated in Figure 4.

## 5. Discussion

In this study of children with different types of cardiomyopathies, we found elevated copeptin levels are associated with worse outcomes in children with cardiomyopathy. This association was significant even after adjusting for the severity of their heart failure. Copeptin levels were found to be a moderately to highly accurate predictor of adverse outcomes in children with cardiomyopathy, with a sensitivity of 86% and specificity of 60% for levels of 25 pg/mL or higher. However, this accuracy was only observed in cases of dilated and restrictive cardiomyopathy, with no predictive value for hypertrophic and LV non-compaction cardiomyopathy. BNP levels were similarly predictive across all types of cardiomyopathies when compared to Copeptin levels. Interestingly, only LVEF was found to be a predictor in cases of hypertrophic and LV non-compaction cardiomyopathy. Overall, these results suggest that copeptin levels may be a useful tool for predicting outcomes in children with certain types of cardiomyopathies. We found that most patients in our group who had end-stage heart failure and were waiting for a heart transplant were in the dilated and restrictive cardiomyopathy group. These patients had severe systolic and diastolic function and were receiving inotropic support. We considered the severity of their heart failure when analyzing the data, but it still might have an impact on the results. This may be why copeptin was a better predictor of adverse outcomes in these patients. Interestingly, copeptin levels were not a reliable predictor in hypertrophic and LV non-compaction cases, which warrants further investigation. Our study sheds light on an area that has not been well explored before, and we recommend conducting a larger multicenter study with comparable heart failure severity in different types of pediatric cardiomyopathies subgroups to further our understanding.

Copeptin is a larger molecular fragment of pre-pro vasopressin (a precursor molecule for AVP and copeptin) secreted in equimolar amounts to Arginine vasopressin (AVP) secreted by hypothalamus neurally transported to the posterior pituitary and released in the bloodstream [18,19]. Excess AVP secretion in response to osmotic and non-osmotic stimuli will cause stimulation of AVP V2 receptors, which leads to free water retention resulting in hyponatremia in congestive heart failure (CHF) [16,28]. The stimulation of atrial and arterial baroreceptors in response to hypotension and volume depletion results in the non-osmotic release of AVP. The predominance of non-osmotic AVP secretion over osmotic AVP release plays a crucial role in developing water imbalance and hyponatremia in CHF [16]. AVP-mediated dilutional hyponatremia, fluid accumulation, and vasoconstriction are associated with worsening heart failure in adults [16]. Hyponatremia in heart failure is an important finding. It has been shown that hyponatremia in heart failure is associated with poor outcomes [16]. In addition, it is also very difficult to treat hyponatremia in patients with heart failure. There has been a significant interest in AVP antagonism in the past years for heart failure with hyponatremia [15,16]. Several AVP antagonists are in different stages of development. Conivaptan is a combined V1 and V2 antagonist, which is approved by US Food and drug administration for use in hyponatremia with or without decompensated heart failure [16,28]. Based on animal and human studies, there is a steady increase in urine output and serum sodium with conivaptan use [16,28,29]. Other promising vasopressin antagonists are tolvaptan, lixivaptan, etc. [14,16,28]. Interestingly, our cohort did not observe significant hyponatremia or decreased serum osmolarity. The above observation might be due to our cohort’s relatively lower severity of heart failure, limited use of diuretic agents, and age-related variation in the neurohormonal milieu. Based on our observation, there is a slight negative correlation between copeptin levels and serum sodium concentration. However, there is no correlation between copeptin levels and serum osmolarity or the patient’s age. This suggests that AVP is mainly released non-osmotically, leading to the release of copeptin in our cohort. These results are similar to the results reported in the adult study [30].

Several studies have shown that copeptin levels can predict outcomes in adults with heart failure [20,21,22,23]. In a study of 268 adult patients with advanced heart failure, Stoiser et al. showed that copeptin has a better ability to predict death compared to BNP [20]. Similarly, in a large cohort of adult patients with all the stages of heart failure, Neuhold et al. showed that copeptin has a similar predictive ability of all-cause mortality compared to BNP, but when both were combined, the predictive ability improved [21]. Maisel et al. showed that increased copeptin levels and hyponatremia are associated with increased 90-day mortality in adult patients with acute heart failure [22]. In this study, the prognostic ability of copeptin increased when it was combined with hyponatremia. Another large prospective study by Pozsonyi et al. showed that copeptin is also a useful long-term prognostic marker to predict all-cause mortality in patients with decreased left ventricular ejection fraction [23]. Based on these observations, copeptin can be used as a short- and long-term prognostic biomarker in the adult heart failure population.

Pediatric data on copeptin level as a predictor of adverse outcomes are limited to few studies, including arterial hypotension in critically ill children, pulmonary hypertension, ventricular dysfunction with vaso-occlusive crises of sickle cell anemia, and pediatric acute heart failure [25,26,31,32,33]. Gaheen et al. showed that copeptin levels were elevated in children with pulmonary hypertension secondary to congenital heart disease [31]. Copeptin levels increased with increased severity of pulmonary hypertension, and higher copeptin levels predicted adverse outcomes [31]. Similarly, Baumann et al. showed that copeptin levels were elevated in critically ill patients with arterial hypotension [25]. A single-center prospective study by Deveci et al. found that copeptin levels were elevated in patients with Sickle cell anemia with vaso-occlusive crisis compared to healthy controls [32]. They concluded that copeptin might help detect vaso-occlusive crisis early in patients with sickle cell disease. Interestingly contrast to adult studies, copeptin levels were not significantly elevated in children with sepsis and septic shock compared to healthy controls [33,34]. Similar to our observations, Amrousy et al. have reported copeptin level as an excellent prognostic value to predict adverse outcomes in pediatric heart failure [26]. However, they did not compare the predictive value of copeptin levels with BNP levels and the echocardiography variables of left ventricular function. Also, their cohort was much younger (mean age of 3.8 years) compared to the median age of 13 years, with a shorter follow-up period of 6 months compared to 2 years in our cohort. Most of their patients (68%) had underlying congenital heart defects as a cause of heart failure, which might explain differences in correlations among variables and a cutoff value of copeptin compared to our study.

It is important to note that our study has certain limitations. Due to a small sample size and diverse patient characteristics, our findings may not apply to a larger population. We could not establish a temporal relationship between copeptin level and heart failure progression due to the lack of serial measurements. Furthermore, we could not consider many confounding factors, including different heart failure treatments, which may have affected our results. However, this is the first study to compare the predictive value of copeptin levels in a cohort of children with cardiomyopathy. Our study provides valuable information to help design future large multi-centered studies.

## 6. Conclusions

Copeptin levels are associated with the severity of heart failure and adverse outcomes in children with cardiomyopathy. Copeptin levels are moderately to highly accurate predictors of adverse outcomes in children with cardiomyopathy. However, this accuracy was only observed in dilated and restrictive cardiomyopathy cases, with limited predictive value for hypertrophic and left ventricular non-compaction cardiomyopathy. BNP levels are similarly predictive across all types of cardiomyopathies. The clinical usefulness of copeptin as a biomarker beyond BNP levels in children with cardiomyopathy needs further investigation.

## Figures and Tables

**Figure 1 children-10-01138-f001:**
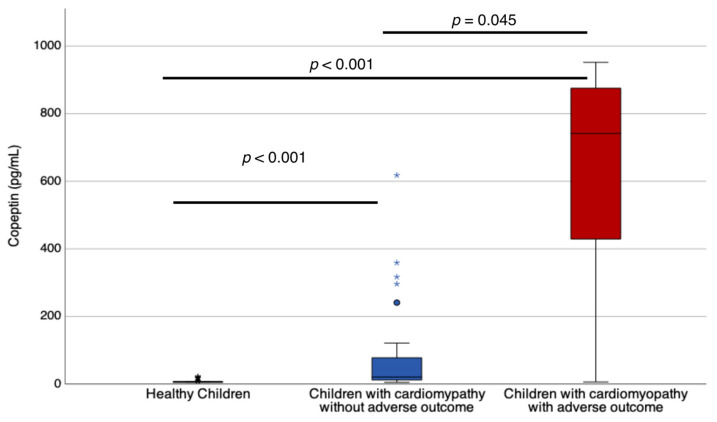
Children with cardiomyopathy and adverse outcomes had significantly higher copeptin levels (median (IQR) of 741.8 pg/mL (351–9.14.1)) than those without adverse outcomes (21 pg/mL (11.9–74.4) and healthy children (6.2 pg/mL (5.8–6.7). The difference was statistically significant, with a *p*-value of 0.045 for children with adverse outcomes and a *p*-value of less than 0.001 for healthy children. * Represents outliers within the groups.

**Figure 2 children-10-01138-f002:**
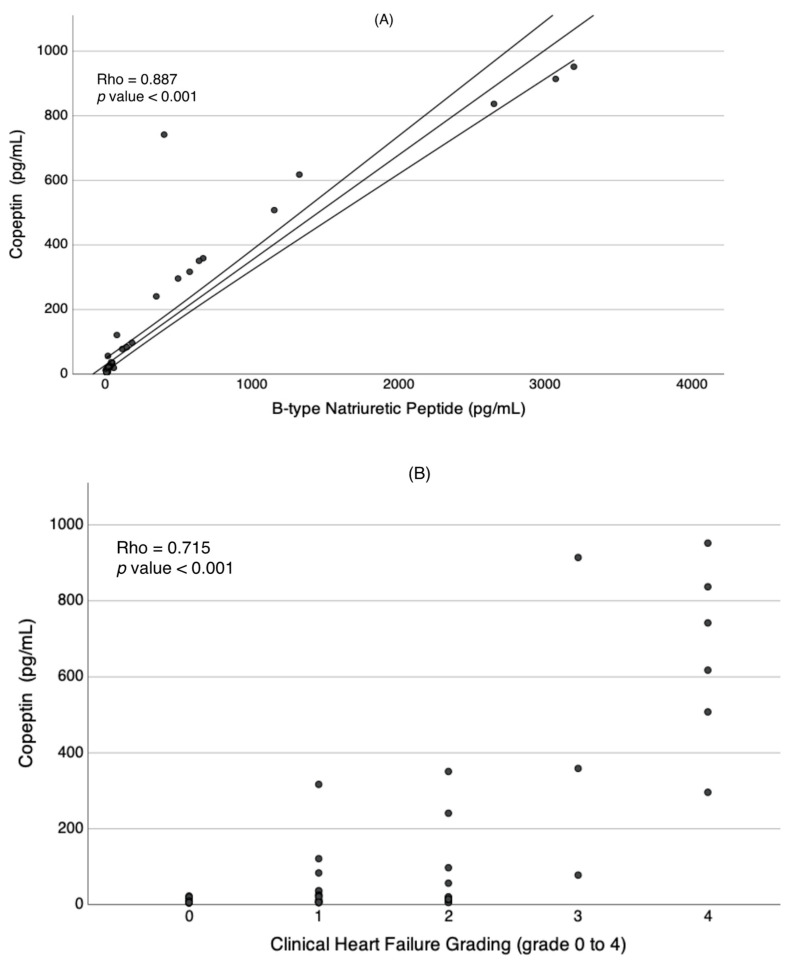
The correlations of copeptin levels with BNP (**A**) and heart failure grading (**B**) are represented by Spearman’s correlation coefficients.

**Figure 3 children-10-01138-f003:**
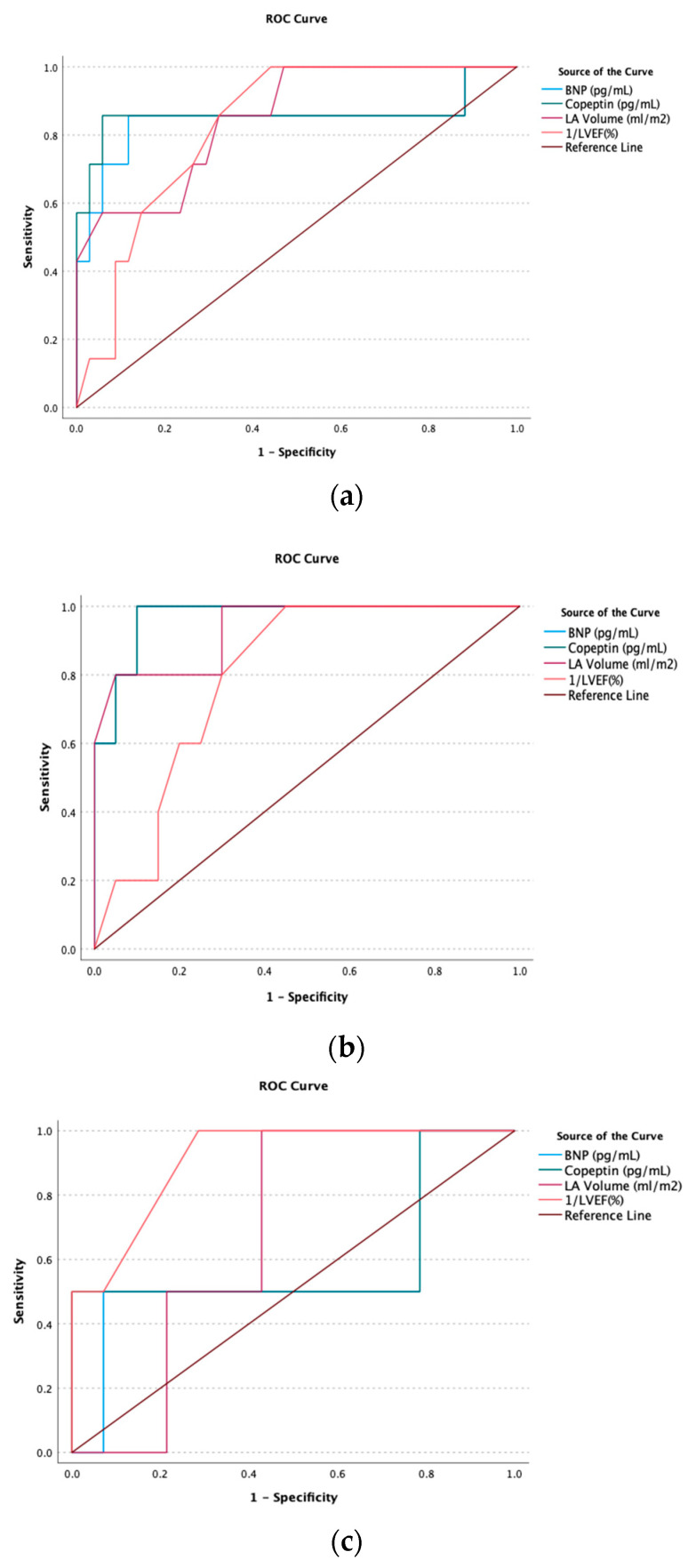
Receiver operating characteristic (ROC) curves that evaluate the ability of copeptin level, B type natriuretic peptide (BNP), left atrial (LA) volume, and left ventricular ejection fraction (LVEF)% predictions to anticipate adverse outcomes in all cases (**a**), cases with dilated and restrictive cardiomyopathy (**b**), and cases with hypertrophic and left ventricular non-compaction cardiomyopathy (**c**).

**Figure 4 children-10-01138-f004:**
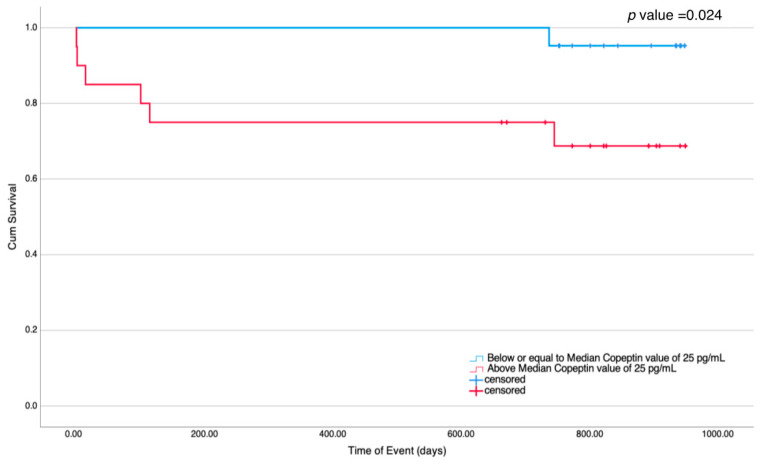
For all cases, using the cutoff median copeptin level of 25 pg/mL, children with cardiomyopathy had a greater likelihood of experiencing adverse outcomes, as shown with the Kaplan–Meier survival curves. *p* value of less than 0.05 was considered significant.

**Table 1 children-10-01138-t001:** The clinical features of children with cardiomyopathy and comparison between those who experienced adverse outcomes and those who did not.

Variable	All Cases *n* = 42	Cases without Adverse Outcomes *n* = 35	Cases with Adverse Outcomes *n* = 7	* p * Value
**Age in years**	13 (2–17)	10 (2–17)	15 (12–17)	0.530
**Weight in kg**	49 (12–72)	40 (11–73)	54 (45–72)	0.552
**Height in cm**	155 (87–171)	152 (83–168)	169 (130–180)	0.181
**Female gender**	17 (40.4)	16 (38)	1 (14.2)	0.222
**Race**				0.578
**African-American** **Caucasian** **Hispanic**	21 (50) 16 (38) 4 (10)	16 (46) 14 (40) 4 (11)	5 (71) 2 (29)	
**Other**	1 (2)	1 (3)		
**Cardiomyopathy type**				0.190
**HCM** **LVNC** **DCM** **RCM**	7 (17) 9 (21) 21 (50) 5 (12)	7 (20) 7 (20) 18 (51) 3 (9)	0 (0) 2 (28) 3 (44) 2 (28)	
**NYHA/Ross grading**				0.003
**I** **II** **III** **IV**	21 (50) 12 (28) 3 (7) 6 (15)	20 (57.1) 11 (31.4) 2 (5.7) 2 (5.7)	1 (14.2) 1 (14.2) 1 (14.2) 4 (57.4)	
**LVEF in %**	50 (40–55)	55 (40–55)	30 (20–45)	0.004
**LA Volume(mL/m^2^)**	25 (15–35)	23 (15–33)	55 (28–68)	0.003
**LVEDD (mm)**	43 (38–54)	43 (37.5–50.5)	56 (39–68)	0.137
**Copeptin in pg/mL**	24.6 (12.3–301)	21.3 (12.3–83.7)	741.8 (351–914)	0.002
**BNP in pg/mL**	32.8 (10.8–425.7)	23.8 (10.8–116.7)	1153 (402–3072)	0.003
**Serum Sodium in mmol/L**	140 (137–142)	140 (139–142.0)	139 (133–140)	0.151
**Serum Creatinine in mg/dL**	0.6 (0.3–0.7)	0.5 (0.3–0.7)	0.7 (0.7–0.80)	0.226
**Serum Osmolarity in mOsmol/L**	296 (292–302)	296 (294–302)	297 (283–298)	0.440

DCM: Dilated cardiomyopathy, LVNC: Left ventricular non-compaction, Cardiomyopathy RCM: Restrictive cardiomyopathy, HCM: Hypertrophic cardiomyopathy, NYHA: New York Heart Association, BNP: B-type natriuretic peptide, LVEF: Left ventricular ejection fraction, and LVEDD: Left ventricular end-diastolic dimension. *p* value of less than 0.05 was considered significant.

**Table 2 children-10-01138-t002:** Correlation of copeptin levels with clinical and laboratory variables by Spearman’s correlation coefficients.

Variable	Spearman’s Rho	95% Confidence Interval	*p* Value
Age in years	0.370	−0.200, 0.269	0.758
HF grading	0.715	0.517, 0.800	<0.001
Serum Sodium (mmol/L)	−0.332	−0.531, −0.098	0.005
Serum Osmolarity (mOsmol/L)	0.103	−0.42, 0.337	0.394
BNP (pg/mL)	0.887	0.824, 0.929	<0.001
LVEF (%)	−0.442	−0.665, −0.146	0.004
LA Volume (mL/m^2^)	0.600	0.351, 0.770	<0.001
LVEDD (mm)	−0.111	−0.420, 0.221	0.502

HF: Heart failure, BNP: B-type natriuretic peptide, LVEF: Left ventricular ejection fraction, LA: Left atrial, and LVEDD: Left ventricular end-diastolic dimension. *p* value of less than 0.05 was considered significant.

**Table 3 children-10-01138-t003:** The results of a univariable logistics regression conducted to investigate the association between clinical features and adverse outcomes in children with cardiomyopathy.

Variable	Odds Ratio	95% Confidence Interval	* p * Value
**Age in years**	1.080	0.945–1.233	0.259
**Female gender**	0.222	0.024–2.047	0.184
**DCM/RCM type**	1.875	0.319–11.021	0.487
**HF grading**			0.025
**II**	1.8	0.103–32	0.683
**III**	10	0.437–228	0.149
**IV**	40	2.8–554	0.006
**Copeptin (pg/mL)**	1.007	1.003–1.012	0.002
**BNP (pg/mL)**	1.003	1.000–1.005	0.023
**LVEF (%)**	0.935	0.886–0.986	0.013
**LA volume (ml/m^2^)**	1.093	1.029–1.161	0.004
**LVEDD (mm)**	1.074	0.999–1.154	0.053

DCM: Dilated cardiomyopathy, RCM: Restrictive cardiomyopathy, HF: Heart failure, BNP: B-type natriuretic peptide, LVEF: Left ventricular ejection fraction, LA: Left atrial, and LVEDD: Left ventricular end-diastolic dimension. *p* value of less than 0.05 was considered significant.

## Data Availability

All data generated or analyzed during this study are included in this article. Further inquiries can be directed to the corresponding author.

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
