# Peer review of "Elevated Copeptin Levels Are Associated with Heart Failure Severity and Adverse Outcomes in Children with Cardiomyopathy"

_children, 2023, doi:10.3390/children10071138_

Round 1

Reviewer 1 Report

This article deepens our understanding of Copeptin as a predictor of heart failure severity and adverse outcomes due to cardiomyopathy, especially as the accuracy of the prediction varies among different types of cardiomyopathy. Because Copeptin as a predictor of heart failure severity and adverse outcomes has been studied well in both adults and children, my major concern is that the authors should highlight cardiomyopathy in terms of innovation, how are the characteristics of heart failure caused by cardiomyopathy different from those caused by other diseases? In particular, why Copeptin as a predictor of heart failure severity and adverse outcomes is limited to heart failure caused by dilated and restrictive cardiomyopathy and not hypertrophic and LV non-compaction cardiomyopathy, the authors should also focus on these differencesin the discussion section.

Minor

1.Please moved Fig legends below Fig.

2. The image resolution of Fig.2 is insufficient. 

This article deepens our understanding of Copeptin as a predictor of heart failure severity and adverse outcomes due to cardiomyopathy, especially as the accuracy of the prediction varies among different types of cardiomyopathy. Because Copeptin as a predictor of heart failure severity and adverse outcomes has been studied well in both adults and children, my major concern is that the authors should highlight cardiomyopathy in terms of innovation, how are the characteristics of heart failure caused by cardiomyopathy different from those caused by other diseases? In particular, why Copeptin as a predictor of heart failure severity and adverse outcomes is limited to heart failure caused by dilated and restrictive cardiomyopathy and not hypertrophic and LV non-compaction cardiomyopathy, the authors should also focus on these differencesin the discussion section.

Minor

1.Please moved Fig legends below Fig.

2. The image resolution of Fig.2 is insufficient. 

Author Response

This article deepens our understanding of Copeptin as a predictor of heart failure severity and adverse outcomes due to cardiomyopathy, especially as the accuracy of the prediction varies among different types of cardiomyopathies. Because Copeptin as a predictor of heart failure severity and adverse outcomes has been studied well in both adults and children, my major concern is that the authors should highlight cardiomyopathy in terms of innovation, how are the characteristics of heart failure caused by cardiomyopathy different from those caused by other diseases? In particular, why Copeptin as a predictor of heart failure severity and adverse outcomes is limited to heart failure caused by dilated and restrictive cardiomyopathy and not hypertrophic and LV non-compaction cardiomyopathy, the authors should also focus on these differences in the discussion section.

Author’s response

Thank you for sharing your insightful comment and suggestion. We agree that copeptin's predictive ability has been well-established in both adult and pediatric heart failure and has proven to be a reliable indicator of heart failure severity and negative outcomes. However, our study focused on its usefulness in predicting pediatric cardiomyopathies, a significant cause of heart failure in children. Unlike heart failure caused by structural heart defects, arrhythmias, sepsis, and other factors, cardiomyopathies are progressive conditions that cannot be reversed by treating the underlying cause. Instead, they inevitably lead to end-stage heart failure and often require heart transplantation. Therefore, close monitoring of these patients is crucial, and biomarkers like copeptin can aid in early detection of severity and negative outcomes, allowing for timely evaluation for heart transplantation. We have included this unique differentiation of the heart failure secondary to cardiomyopathies compared to other etiologies in paragraph number 2, line 46-58.

As you mentioned earlier, our research found that copeptin is more effective in predicting adverse events in dilated and restrictive cardiomyopathies than in hypertrophic and LV non-compaction cardiomyopathies. The outcomes for dilated and restrictive cardiomyopathies are generally worse than other forms of cardiomyopathies. In our study, the majority of patients with end-stage heart failure who required inotropic support were in the dilated and restrictive cardiomyopathies group. These patients were more likely to experience adverse outcomes. We attempted to account for the severity of clinical heart failure using multivariable analysis, but there still may be some confounding factors. Since copeptin hasn't been extensively studied in various forms of cardiomyopathies, our results provide an exciting opportunity for future large-scale studies to investigate cases with similar heart failure severity in different subgroups of pediatric cardiomyopathies.   We have included this description in paragraph number 1, lines 279-289, in the discussion section.

  1. Please move Fig legends below Fig.

We moved Figure legends below the Figures.

  1. The image resolution of Fig 2 is insufficient. 

  We replaced Fig 2 with a better resolution Figure.

Reviewer 2 Report

Karki et al have executed and presented a well-thought study looking into Copeptin as a marker for heart failure severity and outcomes in children with cardiomyopathies. The authors acknowledge that there is pediatric data describing copeptin as a predictor of adverse outcomes in other pediatric pathologies with heart failure. However, this study is certainly unique at this point in time since it looks at subset of children with cardiomyopathy related heart failure. This is certainly an interesting and important topic as it may have an impact on heart failure therapies in the future.

The introduction describes the Copeptin molecule and its pathophysiology in a succinct and understandable manner. The study does not have a clearly stated hypothesis, but the authors are clear and concise in describing their primary endpoints and the descriptive goals of this exploratory study. The methodology and results are adequately described.

The discussion is relevant with appropriate references. The authors have been mindful of recognizing the limitations of their study and it appears that they have made an effort to not over-reach with their conclusions.

Some minor feedback for improvement:

1.  It would be nice to know the duration of enrollment and if they were any calculations done to estimate an appropriate sample size. If not, the authors should clarify that this was a “convenience sample”.

2.  Apart from possibly guiding heart failure treatment in the future, Copeptin may also be used for heart failure monitoring as BNP is used currently. It would be interesting to know the financial cost of obtaining a Copeptin value vs a BNP value.

There are a few minor grammar/syntax issues but overall, the manuscript is well written and easy to read and understand.

Author Response

Karki et al have executed and presented a well-thought study looking into Copeptin as a marker for heart failure severity and outcomes in children with cardiomyopathies. The authors acknowledge that there is pediatric data describing copeptin as a predictor of adverse outcomes in other pediatric pathologies with heart failure. However, this study is certainly unique at this point in time since it looks at subset of children with cardiomyopathy related heart failure. This is certainly an interesting and important topic as it may have an impact on heart failure therapies in the future.

The introduction describes the Copeptin molecule and its pathophysiology in a succinct and understandable manner. The study does not have a clearly stated hypothesis, but the authors are clear and concise in describing their primary endpoints and the descriptive goals of this exploratory study. The methodology and results are adequately described.

The discussion is relevant with appropriate references. The authors have been mindful of recognizing the limitations of their study and it appears that they have made an effort to not over-reach with their conclusions.

Some minor feedback for improvement:

  1. It would be nice to know the duration of enrollment and if they were any calculations done to estimate an appropriate sample size. If not, the authors should clarify that this was a “convenience sample”.

 Author’s response:

Thank you for this question. The duration of enrollment was about one year from May 24,2017 to May 09,2019. We did the sample size calculation based on the effect size of the previous studies. We determined the sample size of 30 in each group, at alpha 0.05 with P > 0.8, to detect the 25% mean difference in copeptin level between the cardiomyopathy and the control groups. We have added this in the manuscript in the statistical analysis section lines 146-148.

  1. Apart from possibly guiding heart failure treatment in the future, Copeptin may also be used for heart failure monitoring as BNP is used currently. It would be interesting to know the financial cost of obtaining a Copeptin value vs a BNP value.

       Author’s response:

  Thank you so much for bringing up this essential point of cost-effectiveness. It costs about $30 to test one sample of copeptin in our university research laboratory. The cost was similar to the BNP assay. Our study was not designed to perform a cost-effectiveness analysis as copeptin levels were never used clinically. We have included this in the methods section of the manuscript in paragraph number 2, lines 135-136.

Round 2

Reviewer 1 Report

Although the authors did not fully adress my concern why Why Copeptin is Specific in Predicting heart failure caused by Cardiomyopathy, there answers are acceptable. I have no further commnets.